# Use of a Vaginally Administered Gel Containing the GnRH Agonist Triptorelin and a Single, Fixed-Time Artificial Insemination in Pigs under Commercial Conditions: Productive and Economic Impacts

**DOI:** 10.3390/ani14182673

**Published:** 2024-09-13

**Authors:** Sara Crespo, Joaquín Gadea

**Affiliations:** 1CEFUSA, 30840 Murcia, Spain; sara.crespo@cefusa.com; 2Department of Physiology, Faculty of Veterinary Science, University of Murcia, Campus Espinardo, 30100 Murcia, Spain

**Keywords:** porcine reproduction, fertility, reproductive management, artificial insemination, reproductive outcomes

## Abstract

**Simple Summary:**

This study addresses the use of triptorelin acetate, a GnRH agonist, to synchronize ovulation in post-weaning sows for timed artificial insemination. This study was designed to evaluate the efficacy and economic impact of this approach under commercial conditions. The results showed that triptorelin effectively induced estrus and synchronized ovulation when administered 96 h post-weaning, particularly in sows with clear signs of estrus. This resulted in comparable reproductive outcomes with fewer inseminations and significant economic benefits. However, sows without clear signs of estrus or with a delayed onset of estrus had poorer outcomes. The results of this study highlight the potential of triptorelin to improve reproductive efficiency and economic performance in swine production, but suggest that careful monitoring and individualized management may be necessary for sows with delayed or equivocal estrus signs. These findings provide valuable insights for optimizing reproductive strategies in commercial swine production.

**Abstract:**

Fixed-time artificial insemination is an important technique in swine production that can improve reproductive efficiency and meat production quality through making better use of the genetic potential of breeding males and reducing the costs associated with double or multiple inseminations. Our goal was to evaluate the vaginal application of the GnRH agonist triptorelin acetate to synchronize ovulation in post-weaning sows and facilitate the implementation of a single, fixed-time insemination. In the first experiment, the efficacy of treatment with triptorelin in animals with or without signs of estrus was analyzed using a single insemination at a fixed time, compared to a control group following the standard insemination strategy. The farrowing rate was comparable between the triptorelin and control groups (100 vs. 87.50%), but triptorelin treatment without estrus had a lower rate (50%). Litter size did not differ between the groups. Estradiol and progesterone levels at 96 and 120 h post-weaning were similar in the control and triptorelin groups (*p* > 0.05). These results suggest that triptorelin has the potential to synchronize ovulation in pigs without affecting post-weaning hormonal profiles. In a second experiment, the objective was to evaluate the productive and economic impact of implementing a treatment with triptorelin acetate 96 h after weaning, compared to the standard insemination protocol. Sows were grouped according to treatment (control vs. triptorelin) and estrus onset (≤5 days and >5 days after weaning, which was considered late estrus). The farrowing rate was lower in the late-estrus control group than in the control and triptorelin groups, and similar to that in the late-estrus triptorelin group. No differences were found in litter size and live or dead piglets born (*p* > 0.05). We developed an estimation model to assess the cost/benefit of intravaginal triptorelin administration at 96 h post-weaning. The overall result was that the use of triptorelin increased the financial benefit per inseminated sow by EUR 15–20. This improvement was mainly related to an increase in the reproductive performance of the treated sows compared to the control sows and the reduction in the number of inseminations per sow. These results highlight the potential of triptorelin to optimize reproductive management in pigs, improving efficiency and economic viability.

## 1. Introduction

Fixed-time artificial insemination is an important technique in swine production that can improve reproductive efficiency and meat production quality through making better use of the genetic potential of breeding males and reducing the costs associated with double or multiple inseminations [1].

An important aspect of the implementation of fixed-time artificial insemination is the correct synchronization between ovulation and time of insemination. In pigs, there is high variability in the time between weaning and the onset of estrus and/or ovulation [2,3,4]. Various hormonal protocols, including those using GnRH analogues, have been used to synchronize the timing of ovulation (reviewed in [1,5]).

Triptorelin is a GnRH analog that binds to receptors in the pituitary gland and stimulates the secretion of the hormones LH and FSH. This compound was first studied by Dr. Andrew Schally [6], who was awarded the 1977 Nobel Prize in Medicine “for their discoveries concerning the peptide hormone production of the brain” [7]. In the porcine species, several studies have evaluated the vaginal application of triptorelin acetate to synchronize post-weaning sows and facilitate a single, fixed-time insemination [8,9]. Preliminary studies on the use of triptorelin defined the gel viscosity conditions for vaginal administration [8] and determined that a dose of 200 µg administered 96 h after weaning was the most effective treatment for inducing ovulation [10]. A single AI given 22–26 h after the triptorelin treatment provided the best results [11], with ovulation occurring 40–48 h after treatment [8,11].

In recent years, studies have been conducted under different conditions and in various production systems to evaluate the feasibility of implementing a fixed-time insemination system combined with the use of triptorelin in sows to effectively synchronize ovulation [12,13,14,15]. Building on this research, our goal was to evaluate the applicability of this method on a large scale under commercial conditions and to study the protocol in a large number of animals. In our first experiment, the aim was to analyze the efficacy of treating animals (with or without signs of estrus) with the GnRH agonist triptorelin, followed by a single insemination at a fixed time and compared to a control group following the standard insemination strategy. A second experiment was designed to evaluate the effect of the day of estrus on the reproductive and economic effects of triptorelin, compared to the standard insemination protocol. Finally, we developed a model to estimate the economic and productive impacts of applying this methodology.

## 2. Material and Methods

### 2.1. Animals

The study was conducted at a commercial, breed-to-wean farm with a herd of 750 sows (Landrace × Large White) crossed with semen doses from Duroc boars, located in the Region of Murcia in southeastern Spain. After weaning, the sows were housed in individual crates with slatted floors for estrus detection and insemination. At day 28 of gestation, they were moved into pens in groups of 24 sows. Approximately 1 week before the expected farrowing date, the sows were moved to farrowing rooms in individual crates. Information on parity and the duration of lactation was recorded for each sow, and sows with parity of 2–8 were maintained in lactation for an average of 26 days. Sows were selected for the study on the basis of their health status to ensure that only healthy animals without a history of reproductive failure were included, thus minimizing potential confounding factors. Sows from each batch were randomly assigned to either the control or experimental group to ensure that any variation in results was due to experimental conditions rather than selection bias. One group of sows was treated with an intravaginal application of gel containing the GnRH agonist triptorelin (200 µg triptorelin) (2 mL OvuGel^®^, Vetoquinol, Madrid, Spain) 96 h after weaning (triptorelin group), while other sows received no treatment (control group).

### 2.2. Estrus Detection and Insemination

Estrus signs were assessed daily, starting immediately after weaning. This process involved placing a mature and active boar in front of the sows to facilitate interaction and stimulate the display of estrus behaviors. The boar remained in front of the sows for a sufficient period of time to allow for proper observation of these signs. A back pressure test was performed on each sow to determine whether the immobilization response occurred. All assessments were supervised by experienced and trained personnel to ensure accuracy and consistency in the identification of estrus. The first day of expression of estrus signs was recorded, and the day and number of inseminations were used to calculate the interval between weaning and insemination (days).

All sows were inseminated with semen from the same group of boars. The insemination dose consisted of 2.5 × 10^9^ total spermatozoa in a total volume of 60 mL and was applied by experienced technicians using a post-cervical artificial insemination catheter. The insemination protocol was adapted for each experiment, as described in the Experimental Design Section.

At 22–25 days after the first insemination, the presence of a pregnancy was determined with an ultrasound scan. The fertility rate, farrowing rate, and litter size were calculated by recording the number of piglets born: total, dead, and alive.

### 2.3. Hormone Analyses

The plasma levels of progesterone (P4. ng/mL) and estradiol (E pg/mL) were evaluated in sows 96 h after weaning and showing signs of estrus, and again 24 h later. Blood samples were collected by direct venipuncture of the jugular vein using lithium heparin collection tubes (BD Vacutainer^®^, BD, Madrid, Spain) and centrifuged at 1500× *g* for 15 min. Plasma was immediately separated and stored at −20 °C until hormone quantification.

Estradiol and progesterone were quantified using an electrochemiluminescence immunoassay (Elecsys^®^ Estradiol III and Progesterone Assays, Olathe, KS, USA), using a Cobas e 411 analyzer (Roche Diagnostics, Barcelona, Spain).

### 2.4. Experimental Design

#### 2.4.1. Experiment 1: Use of Triptorelin in Sows 96 h after Weaning, with Sign of Estrus

A group of 55 sows (parity 2–8) were weaned. Signs of estrus were checked for 96 h after weaning. Thirty-three sows found to be in open estrus were treated with triptorelin, then inseminated with a single dose of semen 22–24 h later (triptorelin group). On the other hand, 16 sows in open estrus did not receive any treatment and were inseminated as usual on the farm with two inseminations, at 0 and 24 h (control group). Finally, a small group of 6 sows without signs of estrus were treated with triptorelin and inseminated 22–24 h after the treatment (triptorelin, no estrus group) (Table 1).

Plasma levels of progesterone (P4. ng/mL) and estradiol (E pg/mL) were evaluated on the day of estrus detection (96 h after weaning) and 24 h later.

For each experimental group, parity, lactation length, gestation length, farrowing rate, total piglets born, and live and stillborn piglets per litter were evaluated.

#### 2.4.2. Experiment 2: Effect of the Timing of Estrus Onset on the Reproductive and Economic Impact of Triptorelin Treatment

A total of 857 sows (parity 2–8) were used in this study. One group of sows (n = 442) was treated vaginally with triptorelin 96 h after weaning (triptorelin group), while 415 sows received no treatment (control group). The sows in each group were divided into two subgroups according to the onset of estrus. Triptorelin-treated sows that showed signs of estrus before 120 h after weaning were inseminated with a single dose of semen at 120 h post-weaning (n = 418). Those sows that showed signs of estrus later than 120 h after weaning were inseminated at 0 and 24 h after the onset of estrus and formed the late-estrus triptorelin group (n = 24) (Table 2).

All control sows were inseminated twice (at 0 and 24 h after the onset of estrus) or three times (with an additional semen dose at 48 h if they were in estrus), according to the usual protocol on the farm. The control sows were divided into two subgroups of 390 and 25 sows, according to the day of onset of estrus, either before or after 120 h after weaning, respectively.

For each experimental group, parity, lactation length, days to estrus onset after weaning, weaning insemination interval (days), number of inseminations per sow, pregnancy rate, gestation length, farrowing rate, day of week of farrowing, total piglets born, and live and stillborn piglets per litter were evaluated.

#### 2.4.3. Experiment 3: Economic Impact Estimation

A cost–benefit model was developed based on the reproductive data from experiment 2 (farrowing rate, live piglets born, number of AIs) for each experimental group. The mean number of viable piglets born per inseminated sow (farrowing rate × live piglets born) was calculated. To estimate the number of piglets that would complete the growing phase and become 20 kg commercial piglets, mortality rates of 14% during lactation and 7% after weaning were used for all groups. However, based on previous studies, the mortality rate during lactation ranges from 10 to 25% of live-born piglets [16,17,18], and the mortality rate after weaning ranges from 4.1 to 7.5% (reviewed by the authors in [19]).

To calculate the income from the sale of 20 kg commercial piglets, it was necessary to evaluate the prices in the EU during the last 5 years. According to the Directorate-General for Agriculture and Rural Development (European Commission), the prices for piglets in Spain and the EU + UK were as shown in Figure 1.

### 2.5. Calculation of AI Costs

In calculating the costs associated with AI, we must first consider the cost of heat detection in the sow, which includes the cost of personnel and the use of a boar. In this case, the costs associated with the heat detection process were the same for all groups because it was necessary to check for estrus in both the triptorelin and control groups.

The cost of the insemination itself includes the cost of the personnel applying the dose of semen, the cost of the dose of semen, and the cost of the material necessary for its application: cleaning of the sow, insemination catheter, and additional material. In this sense, different authors have estimated these costs under different scenarios. The authors of [20] estimated the cost of AI at USD 10.93/insemination, including the sperm dose (USD 7) and the cost of the procedure (operator time, catheter, and other materials; USD 3.93). Alternatively, the authors of [21] calculated the cost at USD 7.17/insemination, with a seminal dose cost of USD 6.00, a catheter cost of USD 0.17, and an average labor cost of USD 1.00 per sow.

In our production conditions in Europe, the average cost of the semen dose was EUR 4–6/dose, and the costs associated with insemination mainly depended on the cost of labor, which in Europe ranged from EUR 15.5/hour in Spain to EUR 27.9/hour in the Netherlands [22]. The time required for insemination is typically close to 5 min for cervical insemination and 2.5 min for post-cervical insemination [23]; however, under our conditions, the estimated time for post-cervical insemination was closer to 1.5 min. The cost of the catheter typically ranges from EUR 0.15 for cervical insemination to EUR 0.60 for post-cervical insemination [24]. Considering all the previous information, to explore the application under different scenarios, we simulated cost–benefit models for the fixed-time protocol when the total cost per insemination was EUR 6, 8, 10, and/or 12. These different insemination costs covered all possible situations in European pig production.

Additionally, we calculated the cost of the application of triptorelin, which included the cost of the compound (EUR 5–5.50/sow) and the time to clean the vulva and apply the treatment (0.5–1 min/sow, EUR 15.5–27.9/hour), with an estimated cost in the range of EUR 0.129–0.465/sow. Thus, the total cost of the treatment could range from EUR 5.13 to EUR 5.97 per sow according to the different scenarios. To facilitate the estimation of the economic impact, we used a mean cost of EUR 5.5 per sow.

### 2.6. Statistical Analysis

The results are expressed as the mean ± SEM. Normality of samples was assessed using the Shapiro–Wilk or Kolmogorov–Smirnov tests, depending on the sample size. Because the parameters did not follow a normal distribution, they were analyzed using the nonparametric Kruskal–Wallis test, which is a nonparametric analog of a one-factor analysis of variance with treatment groups as the main variable. When the Kruskal–Wallis test showed a significant effect, all pairwise multiple comparisons were evaluated. Differences were considered statistically significant at *p* < 0.05. Data were processed using the IBM SPSS software (version 28.0.1.1), and graphs of the data were generated using the Orange data mining software (version 3.36.2).

In this study, the Kruskal–Wallis test was chosen as the primary method of analysis because it is a robust nonparametric alternative to the one-way ANOVA, particularly suitable for data that do not follow a normal distribution. This test is ideal for comparing the medians of multiple groups when the assumptions of normality and homogeneity of variances are not met. Upon finding significant differences with the Kruskal–Wallis test, we conducted post-hoc pairwise comparisons to identify which specific groups differed from each other, ensuring that our findings were statistically sound and accurately reflected the differences observed between groups.

A power analysis, with a desired power of 80% and an alpha level of 0.05, was performed prior to the study to determine the appropriate sample size needed to detect significant differences between groups. This calculation was based on the preliminary information obtained from experiment 1 (under the same experimental conditions) for two main parameters, such as farrowing rate and total number of piglets born. The sample size selected was sufficient to ensure that the study had adequate power to detect meaningful differences between the control and experimental groups.

## 3. Results

### 3.1. Experiment 1: Use of Triptorelin in Sows 96 h after Weaning with Sign of Estrus

The experimental groups were homogeneous in terms of previous lactation duration (Table 3, *p* = 0.664). Regarding the mean parity, no differences were found between the triptorelin and control groups, but that of the triptorelin without estrus group was lower than the others (Table 4, *p* < 0.05). Farrowing was similar between the triptorelin and control groups (100 vs. 87.50%), but the triptorelin without estrus group had a lower farrowing rate (50%, *p* < 0.001). There were no differences in litter size and gestation length between the three groups (Table 4, *p* > 0.05).

At 96 h and 120 h after weaning, estradiol and progesterone levels were similar between the control and triptorelin groups. In both groups, estradiol levels decreased and blood progesterone levels increased 24 h after the onset of estrus, resulting in a decrease in the estradiol/progesterone ratio (Table 5, *p* > 0.05).

### 3.2. Experiment 2: Effect of the Timing of Estrus Onset on the Reproductive and Economic Impact of Triptorelin Treatment

The experimental groups were homogeneous in terms of previous lactation length (Table 6, *p* = 0.190). No differences were found between the triptorelin and control groups with respect to mean parity, but that of the late-estrus triptorelin group was lower than the previous groups and similar to the late-estrus control group (Table 6, *p* = 0.009). The late-estrus groups contained a higher proportion of sows with parity of two (triptorelin, late estrus, 37.5%; control, late estrus, 24%) compared to the groups with estrus before 120 h after weaning (control, 11.8%; triptorelin, 14.8%; chi-square *p* = 0.04). These last groups showed a more homogeneous distribution of sows with parity of 2–8 (Figure 2, violin plots, *p* < 0.04).

The day of estrus onset was similar in the triptorelin and control groups but was significantly higher in the late-estrus control group and even higher in the late-estrus triptorelin group (Table 7, *p* < 0.001). According to the different insemination protocols per group, the interval between weaning and the first insemination, and the number of inseminations were significantly different (Table 7, *p* < 0.001). The weaning-to-first-insemination interval was shorter in the control group than in the triptorelin group, and both were shorter than in the late-estrus control and triptorelin groups (Table 7, *p* < 0.001). In the control groups (normal and late estrus), 85.8% (361/421) of the sows were inseminated twice, while 14.3% (60/421) were inseminated three times; therefore, the mean number of inseminations per sow was 2.12–2.14. This was a higher number of semen doses than in the triptorelin groups, which used one and two AIs per sow (Table 7, *p* < 0.001).

There were no differences in pregnancy rates (Table 7, *p* = 0.296). Gestation length was shorter in the triptorelin group than in either control group, whereas the late-estrus triptorelin group showed an intermediate gestation length (Table 7 and Figure 3, *p* < 0.001). The farrowing rate was lower in the late-estrus control group than in the control and triptorelin groups and similar to the late-estrus triptorelin group (Table 7). No differences were found between the groups for litter size and live or dead piglets born (Table 8, *p* > 0.05).

The distribution of farrows across the days of the week was similar in all groups (chi-squared, *p* = 0.107). The proportion of farrows on the weekend was similar in the control (14%) and triptorelin (12%) groups, while this proportion was lower in the late-estrus control (8.3%) and late-estrus triptorelin (5.9%) groups (Figure 3). Interestingly, in the control group, 209/315 (66.35%) of the farrows were concentrated in the first three days of the week while, in the triptorelin group, this proportion increased to 73.47% (252/343, chi-square *p* < 0.05. Figure 4).

### 3.3. Experiment 3: Economic Impact Estimation

An economic model was estimated based on the reproductive results following the use of triptorelin compared to the reproductive results of the control groups. The model showed that the use of triptorelin corresponded to an increase in the number of commercial 20 kg piglets produced per inseminated sow of 0.20 units when estrus was present before 5 days after weaning (10.58 vs. 10.78; Table 9) and an increase of 1.28 units when estrus occurred after 5 days after weaning (7.77 vs. 9.06; Table 9).

The difference in productivity, measured as the number of commercial 20 kg piglets/inseminated sow, was 0.20 piglets (10.78–10.58, Table 9) for the triptorelin vs. control groups and 1.29 piglets (9.06–7.77) for the late-estrus triptorelin vs. control groups. Considering the prices for the sale of commercial 20 kg piglets in Spain during the period of 2019–2023 (average price EUR 54.84), the use of triptorelin was associated with an increase in income per inseminated sow, ranging from EUR 8.57 to EUR 16.49 for sows in normal estrus (mean value EUR 10.97 = 0.20 piglets × EUR 54.84/piglet) and from EUR 54.86 to EUR 105.56 per inseminated sow for sows in late estrus (mean value EUR 70.20 = 1.29 × 54.84; Figure 5 and Table 10).

Using AI cost estimates of EUR 6, 8, 10, or 12 per insemination, and taking an estimated cost of triptorelin treatment of EUR 5.5/sow, the benefit of using triptorelin was a cost reduction ranging from EUR 1.34 to 8.18 per sow, depending on the AI cost, when estrus onset was 5 days after weaning or before. When the onset of the estrus was later than 5 days, the use of triptorelin increased the reproductive cost in the range of EUR 4.06 to 4.78 (Table 10).

Taken together, the increased income from the sale of piglets and the reduction in reproduction costs represented an economic improvement with the use of triptorelin in normal estrus in the range of EUR 12.31–19.15 per treated sow, while in late estrus, the benefit was between EUR 65.42 and EUR 66.14 per treated sow. As the proportion of late-estrus sows was 3.67% (12/327) in the control group and 4.75% (17/358) in the triptorelin group, the overall result was that the use of triptorelin increased the benefit per inseminated sow in the range of EUR 15–20.

## 4. Discussion

In modern swine production, optimizing reproductive efficiency is a cornerstone of a sustainable and profitable business [25]. A key aspect of this optimization is to synchronize ovulation in sows to ensure a more predictable breeding cycle and maximize resource utilization. Among the many methods available, the use of triptorelin, a potent synthetic analogue of gonadotropin-releasing hormone (GnRH), has emerged as a promising tool for synchronizing ovulation in pigs [9,15]. The strategic use of triptorelin could offer several advantages in swine reproduction. First, it facilitates the tighter control of breeding schedules, allowing producers to better coordinate mating and farrowing times. Second, through synchronizing ovulation, triptorelin promotes more uniform litters. This results in improved litter size and consistent piglet quality. In addition, triptorelin can help to optimize the use of AI through ensuring that ovulation occurs within a defined timeframe, improving AI success rates and ultimately increasing genetic progress within swine herds.

In the first experiment, the efficacy of triptorelin application was tested in sows in estrus 96 h after weaning and inseminated with a single dose 22–24 h later. The reproductive results were similar to those obtained with the normal double-insemination system. However, when triptorelin was administered to sows without obvious signs of estrus and a fixed-time insemination system was used, the results were very poor, limiting the use of a strict fixed-time insemination system for application to all sows, as previously reported by the authors in [13,26]. The small sample size for the triptorelin without estrus group is a limitation of the study; however, we believe that despite the small sample size, the study was sufficient to gather valuable information and demonstrate that this protocol may not be effective under commercial conditions.

In this first experiment, we confirmed that the application of the GnRH analog treatment induced a hormonal pattern similar to that of the control group, with a decrease in estriol levels 24 h after the onset of estrus and a slight increase in progesterone levels [8]. The importance of the timing of insemination is related to the need to maintain viable sperm in the fallopian tubes prior to ovulation [27]. This could be achieved by insemination 24 h prior to ovulation, with an optimal time between 8 and 16 h prior to ovulation, as previously reported [28,29]. The use of GnRH analogues facilitates the synchronization of ovulation in most treated sows up to 40–48 h after treatment [8,11,26]. This means that a single insemination 22–24 h after the onset of estrus will be effective, as we observed in this study.

In the second experiment, we evaluated the effect of the day of onset of estrus on the reproductive and economic impact of triptorelin treatment. In these experiments, the animals that showed the first signs of estrus beyond day 5 after weaning were characterized as having lower parity, and there were significantly more of these type of animals of the second parity. These animals with a delayed onset of estrus showed lower reproductive performance, as previously reported [3]. To facilitate a better reproductive result, this type of animal should be monitored and changes should be made in the duration of lactation to allow a return to estrus in better physical condition.

In both experiments, no differences were observed in the farrowing rate (except for the late-estrus control group), the number of live and dead pigs born, or the total number of pigs born per litter between the triptorelin and control groups. These results confirm the technical feasibility of GnRH agonist treatment, as previously reported in several studies [8,30].

It is important to make efficient use of boars in AI protocols; this is achieved through reducing the number of sperm per insemination dose and increasing the number of semen doses per boar. This achievement maximizes the spread of genetic advances in the reproductive nucleus. [31]. In order to make decisions about the implementation of a fixed-time insemination system, it is necessary to accurately evaluate the costs associated with both the reproductive strategies (labor time, materials, and products) [32] and the reproductive outcomes [20,33], as well as possible benefits, such as the reduction in semen doses resulting in a better use of the productive potential of boars with high genetic value [21,33], as has been evaluated previously in beef cattle [34].

An estimation model was developed to evaluate the cost/benefit of intravaginal triptorelin administration at 96 h post-weaning. In the case of sows with an estrus onset earlier than day 5, the economic improvement was in the range of EUR 15–20/treated sow, while in the case of sows with a late-estrus presentation, the economic improvement was higher and close to EUR 65/treated sow. This improvement is mainly related to an increase in the reproductive performance of the treated sows compared to the control sows. This estimation model can also consider that different groups have different lactation mortality rates (which was not measured), or that synchronizing deliveries reduces weekend care costs and reduces mortality in attended deliveries.

Another interesting consequence of the use of this GnRH agonist is the reduction in gestation length variability. In this sense, we observed a shorter gestation and greater farrowing synchrony in animals treated with triptorelin compared to those that underwent multiple AIs, as has also been reported previously [15]. This fact could facilitate the organization of farrowing supervision and assistance during parturition to reduce mortality during the perinatal period [35], which has a great impact on the productive and economic performance of the farm [36].

Our study provided a valuable insight into the use of triptorelin under commercial conditions. The results showed that while triptorelin can effectively induce estrus and synchronize ovulation in sows, its effectiveness varies depending on the timing of estrus onset. Specifically, our results suggest that this protocol may not be optimal for all sows, particularly those without clear signs of estrus or with delayed estrus onset, where results were less favorable.

As an implication of this study, we noted the importance of careful monitoring of sows, especially those with delayed estrus onset. In commercial swine operations, the timing of estrus is critical to optimizing reproductive efficiency. Sows with delayed estrus, often characterized by a later return to estrus after weaning, tend to have lower reproductive performance [4,37]. This can result in smaller litter sizes, lower pregnancy rates, and reduced overall productivity, which directly impacts the economic viability of the farm. Given these challenges, our findings suggest that management strategies for these sows may need to be adjusted. In addition, closer monitoring of estrus signs and a more individualized approach to insemination timing may be beneficial for this subset of sows, in order to ensure that insemination occurs at the optimal time relative to ovulation.

In addition, we suggest that future research should focus on optimizing the use of triptorelin, particularly in sows with delayed estrus or unclear estrus signs [13]. This may require larger sample sizes to provide more robust data and confirm the trends observed in our study. Through addressing these limitations and exploring alternative strategies, future studies could help refine the use of triptorelin, making it a more reliable tool for improving reproductive outcomes in a broader range of sows.

## 5. Conclusions

Triptorelin administration effectively induced estrus and synchronized ovulation, particularly within 96 h of weaning when clear signs of estrus were detected, resulting in a reduction in the number of artificial inseminations needed to produce similar reproductive outcomes. While no significant differences in pregnancy rates or litter size were observed, sows with late estrus onset had lower reproductive performance. However, triptorelin treatment resulted in economic benefits through increasing the income per inseminated sow, particularly in sows with a late-estrus presentation. These results highlight the potential of triptorelin to optimize reproductive management in pigs, improving efficiency and economic viability.

## Figures and Tables

**Figure 1 animals-14-02673-f001:**
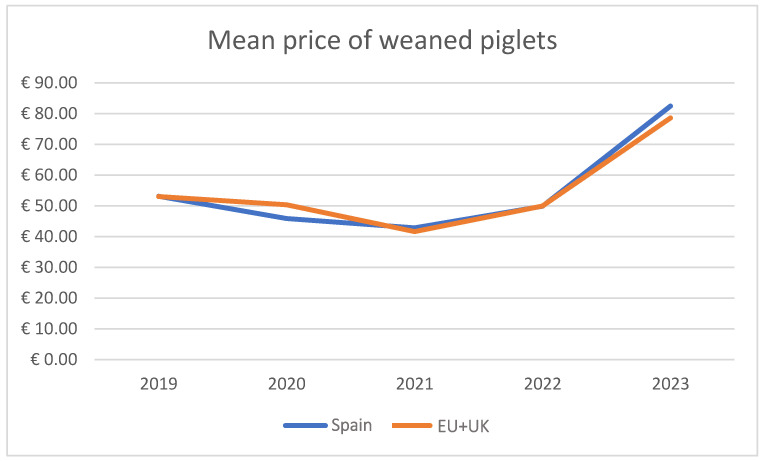
Prices for 20 kg piglets in Spain and the EU+UK. Directorate-General for Agriculture and Rural Development (European Commission).

**Figure 2 animals-14-02673-f002:**
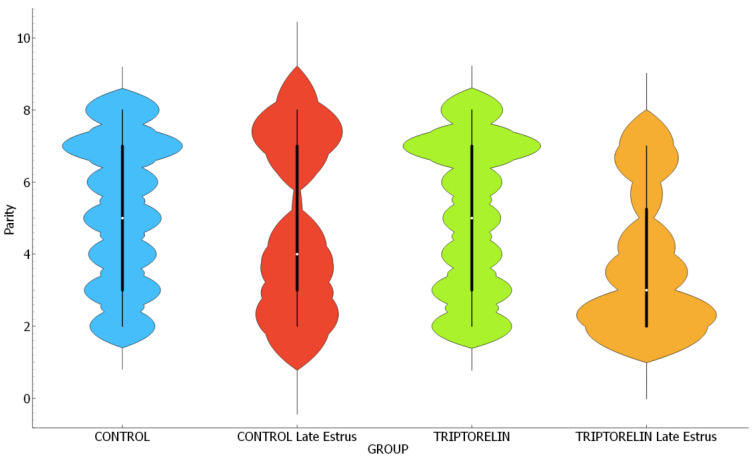
Violin plot of asymmetric distribution of parity of sows from different experimental groups. Chi-square *p* = 0.04.

**Figure 3 animals-14-02673-f003:**
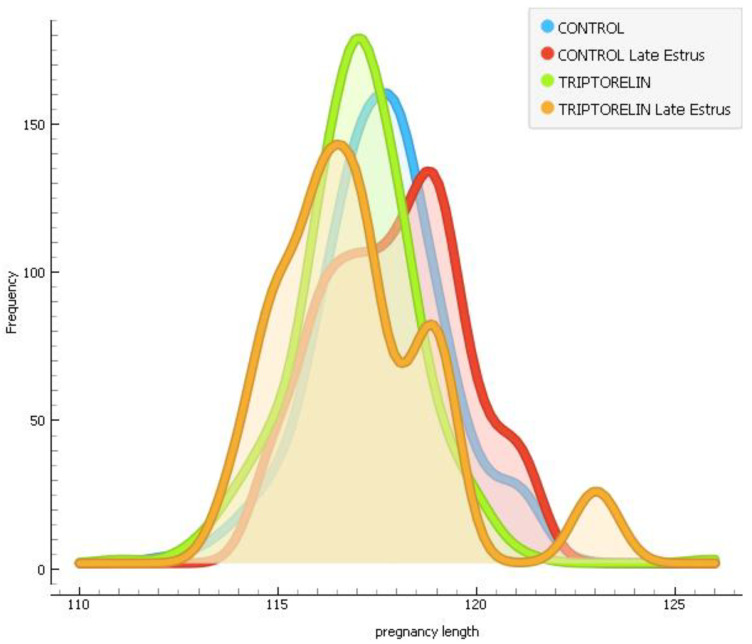
Kernel density distribution of the pregnancy length (in days) of sows from different experimental groups.

**Figure 4 animals-14-02673-f004:**
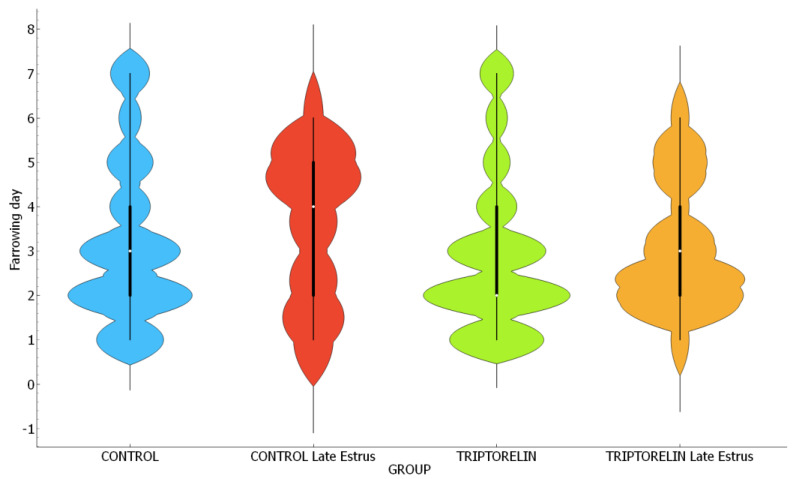
Violin plot of symmetric distribution of farrowing day (Monday, 1; Sunday, 7) of sows from different experimental groups. Chi-square *p* = 0.107.

**Figure 5 animals-14-02673-f005:**
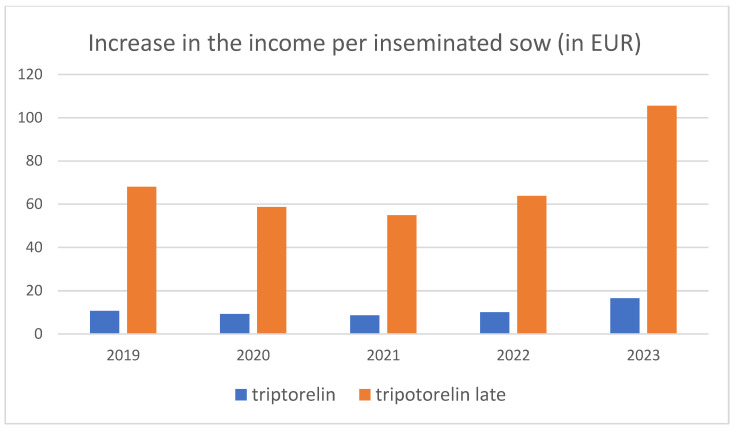
Increase in the income from the sale of commercial 20 kg piglets derived from the use of triptorelin in comparison to control group, in sows with normal onset of estrus before 5 days after weaning or later.

**Table 1 animals-14-02673-t001:** Experimental design of experiment 1. Treatment of the sows after weaning in the different experimental groups.

Group	N	Estrus 96 h Post-Weaning	Triptorelin 96 h Post-Weaning	AI 96 h Post-Weaning	Estrus 120 h Post-Weaning	AI 120 h Post-Weaning
Control	16	+	−	+	+	+
Triptorelin	33	+	+	−	+	+
Triptorelin, no estrus	6	−	+	−	−	+

+: present; −: absent.

**Table 2 animals-14-02673-t002:** Experimental design of experiment 2. Treatment of the sows after weaning in the different experimental groups.

Group	N	Triptorelin 96 h after Weaning	Estrus Onset before 120 h after Weaning	Artificial Insemination
Control	390	No	Yes	2–3 (0 and 24 h, 0.24, 48 h) after onset of sign of estrus
Control, late estrus	25	No	Later	2–3 (0 and 24 h, 0.24, 48 h) after onset of sign of estrus
Triptorelin	418	Yes	Yes	1 (24 h) with sign of estrus
Triptorelin, late estrus	24	Yes	Later	2 (0.24 h) after onset of sign of estrus

**Table 3 animals-14-02673-t003:** Characteristics of sows in the control and triptorelin groups. Parity and previous lactation length in days. Mean ± SEM (n).

	n	Parity	Previous Lactation Length
Control	16	4.13 ± 0.26 ^a^	27.31 ± 1.23
Triptorelin	33	3.88 ± 0.19 ^a^	27.61 ± 1.25
Triptorelin, no estrus	6	3.00 ± 0.26 ^b^	25.67 ± 1.76
*p*-Value		0.062	0.664

^a, b^ A different letter indicates a significant difference among the groups.

**Table 4 animals-14-02673-t004:** Characteristics of sows in the control and triptorelin groups. Mean ± sem (n).

	n	Farrowing Rate (%)	n	Gestation Length (Days)	Live Born	Dead Born	Total Piglets Born
Control	16	87.50 ± 8.54 ^a^	14	117.14 ± 0.29	18.07 ± 0.68	1.14 ± 0.25	19.21 ± 0.67
Triptorelin	33	100 ^a^	33	116.30 ± 0.26	16.30 ± 0.75	1.91 ± 0.52	18.21 ± 0.69
Triptorelin, no estrus	6	50 ± 22.36 ^b^	3	117 ± 1.73	14.33 ± 1.86	1.67 ± 1.67	16.00 ± 1.53
*p*-Value		<0.001		0.220	0.195	0.777	0.269

^a, b^ A different letter indicates a significant difference among the groups.

**Table 5 animals-14-02673-t005:** Estrogen (pg/mL) and progesterone (ng/mL) in blood serum from sows 96 and 120 h after weaning, treated with triptorelin acetate (triptorelin group) or not treated (control). Mean ± SEM.

	Estrogen 96 h after Weaning	Estrogen 120 h after Weaning	Progesterone 96 h after Weaning	Progesterone 120 h after Weaning	Ratio of E/P 96 h after Weaning	Ratio of E/P 120 h after Weaning
Control (n = 15)	21.06 ± 3.38	8.81 ± 1.77	0.16 ± 0.03	0.26 ± 0.06	161.95 ± 25.53	97.98 ± 39.63
Triptorelin (n = 14)	28.12 ± 4.17	7.09 ± 1.90	0.22 ± 0.06	0.33 ± 0.09	206.83 ± 49.29	33 ± 98 ± 12.65
*p*-Value	0.278	0.531	0.306	0.486	0.417	0.207

**Table 6 animals-14-02673-t006:** Characteristics of sows in the control and triptorelin groups. Mean ± SEM.

	Parity (n)	Previous Lactation Length (n)
Control	5.21 ± 0.10 ^a^ (390)	25.96 ± 0.33 (389)
Control, late estrus	4.80 ± 0.47 ^a,b^ (25)	26.56 ± 1.03 (25)
Triptorelin	5.11 ± 0.10 ^a^ (418)	26.11 ± 0.25 (418)
Triptorelin, late estrus	3.79 ± 0.39 ^b^ (24)	27.63 ± 1.14 (24)
*p*-Value	0.009	0.190

^a, b^ A different letter indicates a significant difference among the groups.

**Table 7 animals-14-02673-t007:** Characteristics of sows in the control and triptorelin groups. Mean ± SEM.

	Day of Estrus Onset	Weaning Insemination Interval (Days)	Number of AIs	Pregnancy Rate (%)	Gestation Length	Farrowing Rate
Control	4.50 ± 0.03 ^a^ (388)	4.50 ± 0.03 ^a^ (388)	2.14 ± 0.02 ^a^ (395)	92.04 ± 1.35 (402)	117.60 ± 0.09 ^a^ (315)	85.83 ± 1.82 (367) ^a^
Control, late estrus	6.31 ± 1.21 ^b^ (25)	6.31 ± 1.21 ^b^ (25)	2.12 ± 0.06 ^a^ (25)	88.46 ± 6.39 (25)	117.92 ± 0.51 ^a^ (12)	66.67 ± 11.43 (18) ^b^
Triptorelin	4.52 ± 0.25 ^a^ (421)	5.00 ^c^ (421)	1.00 ^b^ (421)	93.16 ± 1.23 (424)	116.97 ± 0.08 ^b^ (341)	86.40 ± 1.72 (397) ^a^
Triptorelin, late estrus	8.09 ± 0.85 ^c^ (23)	8.09 ± 0.85 ^b^ (23)	2.00 ^a^ (24)	83.33 ± 7.77 (24)	117.00 ± 0.52 ^a,b^ (17)	77.27 ± 9.14 (22) ^a,b^
*p*-Value	<0.001	<0.001	<0.001	0.296	<0.001	0.085

^a, b, c^ A different letter indicates a significant difference among the groups.

**Table 8 animals-14-02673-t008:** Characteristics of sows in the control and triptorelin groups. Mean ± SEM.

	n	Live Born	Dead Born	Total Piglets Born
Control	315	15.42 ± 0.21	1.78 ± 0.08	17.19 ± 0.20
Control, late estrus	12	14.58 ± 1.41	2 ± 0.28	16.58 ± 1.61
Triptorelin	341	15.60 ± 0.22	1.75 ± 0.07	17.36 ± 0.22
Triptorelin, late estrus	17	14.65 ± 0.82	1.94 ± 033	16.59 ± 0.85
*p*-Value		0.602	0.543	0.717

**Table 9 animals-14-02673-t009:** Estimation of commercial 20 kg piglets produced per inseminated sow according to the treatment with or without triptorelin and the timing of estrus onset.

	Farrowing Rate (%)	Alive Born	Alive Born × Farrowing = Alive Born/Inseminated Sow	Weaned/Inseminated Sow (86% Survival)	Commercial 20 kg Piglets/Inseminated Sow (93% Survival)
Control	85.83	15.42	13.23	11.38	10.58
Control, late estrus	66.67	14.58	9.72	8.36	7.77
Triptorelin	86.4	15.6	13.48	11.59	10.78
Triptorelin, late estrus	77.27	14.65	11.32	9.74	9.06

**Table 10 animals-14-02673-t010:** Estimation of the reproductive cost per sow treated with triptorelin in comparison to control group, in sows with normal or late onset of estrus.

Group	No. AIs	Triptorelin Cost (EUR)	Reproductive Cost (EUR 6/AI)	Reproductive Cost (EUR 8/AI)	Reproductive Cost (EUR 10/AI)	Reproductive Cost (EUR 12/AI)	Increased Income (EUR)
Control	2.14	0	12.84	17.12	21.40	25.68	
Control, late estrus	2.12	0	12.72	16.96	21.20	25.44	
Triptorelin	1	5.5	11.5 (−1.34)	13.5 (−3.62)	15.50 (−5.9)	17.5 (−8.18)	+10.97
Triptorelin, late estrus	2	5.5	17.5 (+4.78)	21.5 (+4.54)	25.50 (+4.30)	29.5 (+4.06)	+70.20

## Data Availability

The data supporting the findings of this study are available from the corresponding author upon reasonable request.

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
