# Peer review of "Use of a Vaginally Administered Gel Containing the GnRH Agonist Triptorelin and a Single, Fixed-Time Artificial Insemination in Pigs under Commercial Conditions: Productive and Economic Impacts"

_animals, 2024, doi:10.3390/ani14182673_

Round 1

Reviewer 1 Report

Comments and Suggestions for Authors

The manuscript presents a study on the application of a vaginally administered GnRH agonist, triptorelin, in fixed-time artificial insemination (FTAI) of pigs under commercial conditions. The research assesses both productive and economic impacts, comparing sows treated with triptorelin to control groups under different estrus conditions. The study is divided into two experiments: the first focuses on evaluating the efficacy of triptorelin in synchronizing ovulation, while the second assesses the economic and productive outcomes of this methodology. The findings suggest that triptorelin can improve reproductive efficiency and economic viability in commercial swine production.

The manuscript addresses an important topic in the field of animal reproduction, particularly in swine production, where efficient reproductive management is crucial for economic success. The study is well-structured, with a clear hypothesis and methodology. The experiments are logically designed, and the data is presented in a way that supports the conclusions drawn by the authors.

However, there are some areas where the manuscript could be improved:

  • Language: Please check the manuscript with a native speaker.
  • Literature Review: While the introduction provides a general overview of FTAI and the role of GnRH agonists, it could benefit from a more comprehensive review of recent studies on the use of triptorelin in swine. This would help to better position the current study within the existing body of research.

  • Methodology Clarity: The methodology section is detailed, but some aspects, such as the criteria for selecting sows for the different experimental groups, when did assement of oestrus signs started, when was the boar put infront of the sows and how long, could be more explicitly stated. This would improve reproducibility and help readers better understand the study's design. Furthermore, add the calculation of sample size with the power calculation.

  • Data Analysis: The statistical analysis is appropriate, but the manuscript would benefit from a more in-depth discussion of the statistical methods used, particularly in relation to how differences between groups were tested and interpreted.

  • Discussion and Implications: The discussion effectively summarizes the findings, but it could be strengthened by a deeper exploration of the implications of these results for commercial swine production. Specifically, discussing potential limitations of the study and how they might be addressed in future research would be valuable.

Comments on the Quality of English Language

The whole manuscript requires a language check. Please use the right terminology.

Author Response

Please, see the pdf file.

Reviewer 2 Report

Comments and Suggestions for Authors                       The topic is interesting and important in pig farming. I appreciate your effort, because the research was conducted on a farm and required good organization. Cost calculations are of course variable and different in individual countries. In general, I have no objections because they can be calculated differently. I have a few doubts, which I would like to check and clarify: - Line 125: is (n=415) - should be (442) because 418 + 24 = 442. Both subgroups received triptorelin. - in the control group with late estrus, three-time insemination was used, while in the Triptorelin late estrus group only two-time insemination was used. Did the estrus in these animals last shorter and could it have had an impact on the effectiveness of fertilization? - in the TRIPTORELIN NO ESTRUS group there are only 6 sows (Table 3) or 3 sows (Table 4). Drawing conclusions with such a small number is burdened with error. Therefore, in the discussion of the results, this should be noted. - Line 295-301 (table 10) - the calculated costs should be explained in more detail, e.g. 70.20 EURO (state in brackets how it was calculated)

Author Response

Please, see the pdf file.

Round 2

Reviewer 1 Report

Comments and Suggestions for Authors

The authors have significantly improved the manuscript. Hence, I would like to recommned the manuscript for publication.

Comments on the Quality of English Language

Minor editing of English language required.